

# Functional expression and purification of DoxA, a key cytochrome P450 from *Streptomyces peucetius* ATCC 27952

Liyan Yang[1], Dengfeng Yang[2], Qingyan Wang[1], Juan Li[1], Hong-Liang Li[2] and Lixia Pan[1]

[1] National Engineering Research Center for Non-Food Biorefinery, State Key Laboratory of Non-Food Biomass and Enzyme Technology, Guangxi Academy of Sciences, Nanning, China
[2] Guangxi Key Laboratory of Marine Natural Products and Combinatorial Biosynthesis Chemistry, Guangxi Academy of Sciences, Nanning, China

## ABSTRACT

The antitumor drug doxorubicin is widely used in clinical practice. However, the low yield and high cost of this drug highlight the urgent need for cost-effective processes to rapidly manufacture antitumor drugs at scale. In the biosynthesis pathway, the multi-functional cytochrome P450 enzyme DoxA catalyzes the last three steps of hydroxylation. The final conversion of daunorubicin to doxorubicin is the rate-limiting step. In our work, the DoxA has been expressed with the ferredoxin reductase FDR2 and the ferredoxin FDX1 and purified to homogeneous. The reduced carbon monoxide difference spectroscopy, heme concentration, and enzymatic characteristic were characterized. These studies suggest an approach for engineering *Streptomyces* P450s with functional expression for mechanistic and structural studies.

# INTRODUCTION

Cytochrome P450 (P450) is a ubiquitous heme-dependent enzymes that catalyze multiple reactions through a complex multistep mechanism (*Rudolf et al., 2017*). As the prosthetic group of P450 enzymes, heme is linked to an absolutely conserved cysteine. P450 was named as its reduced state produces a characteristic absorption peak at 450 nm when combined with carbon monoxide (*Klingenberg, 1958*; *Omura & Sato, 1962*). The P450s typically act as monooxygenases, binding dioxygen to their ferrous heme iron and ultimately inserting an atom of oxygen into the substrate, with the other oxygen atom being reduced to water (*Rudolf et al., 2017*). They are well known for their roles in human heterologous detoxification, steroid biosynthesis and drug metabolism (*de Montellano, 2015*), but also play a key role in the biosynthesis of natural products (*Podust & Sherman, 2012*; *de Montellano, 2015*). One key structural feature of P450s is the coordination of the thiolate anion of cysteine to the heme iron as the fifth ligand in the active form (*Sono, Andersson & Dawson, 1982*; *Champion et al., 1982*; *Stern & Peisach, 1974*; *Collman & Sorrell, 1975*; *Sun et al., 2013*). The biologically inactive conformation of a cytochrome P450 protein is usually denoted as the P420 form, which is characterized by a CO bound Soret peak at

Corresponding author
Lixia Pan, panlixia@gxas.cn

420 nm (*Sun et al., 2013*). *Streptomyces* P450s are expressed sometimes as P420 forms, the biologically inactive forms which possess ferrous CO Soret absorption at 420 nm (*Healy et al., 2002*). Therefore, it is crucial to obtain high-level expression and active P450 for its function research.

P450 systems mainly comprise two functional parts: the heme-containing P450 domain, and redox partners (RPs). The heme-containing P450 domain contributes to the binding and transformation of substrate, and the RPs contain redox centers to relay electron equivalents from electron donors, NAD(P)H in most cases, to activate the dioxygen bound to the P450 domain (*Chen et al., 2021*). P450 systems can be classified based on the redox partners required for catalytic activity. In general, a P450 catalytic system includes four components: the substrate, a P450 enzyme for substrate binding and oxidative catalysis, the redox partner(s) that functions as an electron transfer shuttle, and the cofactor, which provides the reducing equivalents (*Li et al., 2020*). Most bacterial P450s belong to the Class I P450 system, which require two RP proteins: an NAD(P)H-dependent ferredoxin reductase (FdR) and ferredoxin (Fdx), and the electron transfer chain is NAD(P)H $\rightarrow$ FAD $\rightarrow$ Fe-S cluster $\rightarrow$ heme (*Hannemann et al., 2007*) (Fig. 1).

Doxorubicin (DXR) is a potent antitumor drug which is the anthracycline drug. In the biosynthesis of DXR, multi-functional Cytochrome P450 enzyme DoxA is responsible for the final three-step hydroxylation (Fig. 2). DoxA has been purified by *Rimal et al. (2015)* but it is mostly in inactive P420 forms (Fig. S1). In order to acquire more active DoxA to study its function and structure, we tried different constructions to express DoxA in this paper. There are six FDXs and seven FDRs in *Streptomyces peucetius*, and the redox partner of DoxA has been identified by *Rimal et al. (2015)*, their study suggested the primary electron-transport pathway of DoxA is NADH $\rightarrow$ FDR2 $\rightarrow$ FDX1 $\rightarrow$ DoxA. Most P450s require redox partner proteins to sequentially transfer two electrons form NAD(P)H to their heme–iron reactive center for dioxygen activation (*Ruettinger & Fulco, 1981*). In this study, FDR2, FDX1 and DoxA of *Streptomyces peucetius* were co-expressed in *E. coli* expression system. The unprecedented high-efficiency and functional expression and purification of DoxA in *E. coli* expression system was realized, and the enzymatic assay of DoxA using daunorubicin (DNR) as the substrate was also performed.

## MATERIAL AND METHODS

### Strains and materials

*E. coli* strains were grown at 37 °C in Luria Bertani (LB) media in both liquid and agar plates supplemented with the appropriate amount of antibiotic. *E. coli* DH5 $\alpha$ was used for recombinant plasmid construction. *E. coli* BL21 Codon plus (DE3) RIL was used as protein expression host. The plasmids pET22b, pET28a, pRSFDuet and pETDuet (Table 1) were used as expression vectors. Antibiotics were added at the following concentrations for *E. coli*: kanamycin (Kan) 50 $\mu$g/ml; ampicillin (Amp) 100 $\mu$g/ml; chloramphenicol (Cm) 25 $\mu$g/ml. The following supplement was added when required: isopropyl- $\beta$-D-thiogalactopyranoside (IPTG) 0.1 mM; $Fe^{2+}$ 0.5 mM, $\sigma$-aminol evulinic acid (ALA) 0.5 mM.

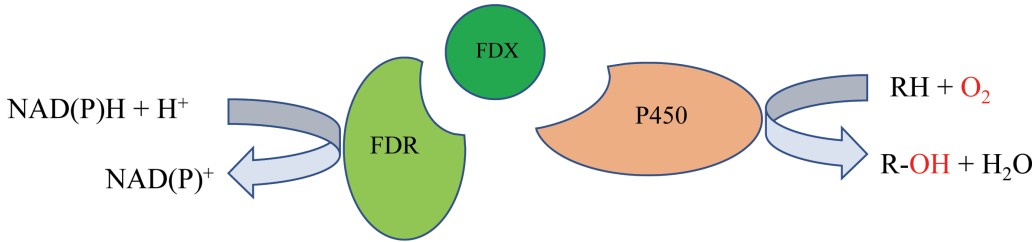

**Figure 1**  **The mechanism of bacterial type I P450-related electron-transport system.**

Daunorubicin → $doxA$ → Doxorubicin

**Figure 2**  **The biosynthetic pathway catalyzed by DoxA.**  The red box is the difference between daunorubicin and doxorubicin.

## Molecular cloning and construction of recombinant plasmids

$doxA$ gene for construction to different expression vector was amplified with primers listed in Table 2. Ferredoxin ($fdx1$) gene and Ferredoxin reductase ($fdr2$) gene were amplified with primer sets fdx1-F/R and fdr2-F/R (Table 2), respectively. Total DNA of *S. peucetius* 27952 strain was used as the PCR template. PCR products were purified with PCR clean-up kit according to manufacturer's description.

The purified $doxA$ fragment was ligated into pRSFDuet that was digested with *Eco* R I and *Hin* d III by ClonExpress II One Step Cloning Kit (Vazyme, China), generating the recombinant plasmid named pEA (Table 1) and its structure is shown in Fig. S2. Other expression plasmids 22bA and 28aA (Table 1) were constructed by the same strategy. The plasmids pEA, 22bA and 28aA were respectively transformed into expression host BL21 Codon plus (DE3) RIL generating the strains pEA/RIL, 22bA /RIL, and 28aA /RIL.

The purified $fdx1$ fragment was ligated into the multiple cloning site (MCS) 2 of pRSFDuet-1 that was digested with *Bgl* II and *Xho* I by ClonExpress II One Step Cloning Kit (Vazyme, China), of the resulting vectors containing $doxA$ gene in the MCS 1 and $fdx1$ gene in the MCS 2, generating the recombinant plasmid named pEAX1 (Table 2). The plasmid pEAX1 was transformed into BL21 Codon plus (DE3) RIL, generating the strain pEAX1/RIL. The purified $fdr2$ fragment was ligated into the MCS 2 of pETDuet that was digested with *Bgl* II and *Xho* I by ClonExpress II One Step Cloning Kit (Vazyme, China),

**Table 1** Bacterial strains and plasmids used in this work.

| Strains or plasmids | Relevant characteristics[a] | Reference or source |
|---|---|---|
| *Escherichia coli* | | |
| DH5 $\alpha$ | $F^-$ $\Phi80lac$ Z $\Delta$M15 $\Delta$(*lac* ZYA-*arg* F) U169 *rec* A1 *end* A1 *hsd* R17($r_k^-$,$m_k^+$) *pho* A *sup* E44 *thi*-1 *gyr* A96 *rel* A1 $\lambda^-$ | Gibco BRL, Life Technologies |
| BL21 Codon plus (DE3) RIL | $F^-$ *ompT* *hsdS* ($r_B^-$ $m_B^{--}$) $dcm^+$ *gal* $\lambda$(DE3) *end* A Hte [*argU ileY leuW* Cam$^r$ ] | Novagen |
| 22bA/RIL | BL21 Codon plus (DE3) RIL harboring 22bA, Amp$^r$, Cam$^r$ | This work |
| 28Aa/RIL | BL21 Codon plus (DE3) RIL harboring 28aA, Kan$^r$, Cam$^r$ | This work |
| pEA/RIL | BL21 Codon plus (DE3) RIL harboring pEA, Kan$^r$, Cam$^r$ | This work |
| pEAX1/RIL | BL21 Codon plus (DE3) RIL harboring pEAX1, Kan$^r$, Cam$^r$ | This work |
| AX1R2/RIL | BL21 Codon plus (DE3) RIL harboring pEAX1 and pER2, Kan$^r$, Amp$^r$, Cam$^r$ | This work |
| Plasmids | | |
| pET22b | Expression vector, C-terminal 6×His-tagged sequences, Amp$^r$ | Novagen |
| pET28a | Expression vector, N-terminal 6×His-tagged sequences, Kan$^r$ | Novagen |
| pRSFDuet | Expression vector which contains two multiple cloning sites (MCS), N-terminal 6×His-tagged sequences, Kan$^r$ | Novagen |
| pETDuet | Expression vector which contains two multiple cloning sites (MCS), N-terminal 6×His-tagged sequences, Amp$^r$ | Novagen |
| 22bA | pET22b containing *doxA* coding region, Amp$^r$ | This work |
| 28aA | pET28a containing *doxA* coding region, Kan$^r$ | This work |
| pEA | pRSFDuet containing *doxA* coding region, Kan$^r$ | This work |
| pEAX1 | pRSFDuet containing *doxA* and *fdx1* coding region, Kan$^r$ | This work |
| pER2 | pETDuet containing *fdr2* coding region, Amp$^r$ | This work |

**Notes.**

[a]Kan$^r$, Amp$^r$ and Cam$^r$ indicate resistance to kanamycin, ampicillin and chloramphenicol, respectively.

**Table 2** Primers used in this study.

| Primer Name | Sequence |
|---|---|
| 22b-doxA-F | taagaaggagatatacatatgGTGAGCGGCGAGGCGCCC |
| 22b-doxA-R | gtggtggtggtggtgctcgagGCGCAGCCAGACGGGCAG |
| 28a-doxA-F | gtgccgcgcggcagccatatgGTGAGCGGCGAGGCGCCC |
| 28a-doxA-R | ctcgagtgcggccgcaagcttTCAGCGCAGCCAGACGGG |
| RSF-doxA-F | ccacagccagggatccgaattcGTGAGCGGCGAGGCGCCC |
| RSF-doxA-R | gcattatgcggccgcaagcttTCAGCGCAGCCAGACGGG |
| fdx1-F | taagaaggagatatacatatgATGACCGTGCAGCACGAGG |
| fdx1-R | ggtttctttaccagactcgagTCACTCCGCGTCCGGGCC |
| fdr2-F | agatatacatatggcagatctGCATCACCATCATCACCACCTTCGCATCGCCGTC |
| fdr2-R | ggtttctttaccagactcgagTCAGCGGGCCGCGTCCGG |

generating the recombinant plasmid named pER2. The plasmid pEAX1 and pER2 were co-transformed into BL21 Codon plus (DE3) RIL, generating the strain AX1R2/RIL.

## Overexpression of *doxA*, *fdx1* and *fdr2* in *E. coli* strains

A single colony of AX1R2/RIL was inoculated in 10 mL LB medium over night at 37 °C as the seed culture. One percent seed were transferred in 500 mL LB medium in 2 L shaking flask. Cells were grown at 37 °C to $OD_{600}$ about 0.6−0.8, then induced by adding IPTG to final concentration of 0.1 mM, ALA and $Fe^{2+}$ were added to 0.5 mM, and the cells were incubated for 24 h at 16 °C. The cell pellets were harvested by centrifugation at 7,000 g for 15 min and resuspended with buffer containing 50 mM Tris, 300 mM NaCl, 20% glycerol, pH 7.5. Finally, the cell pellets were lysed by ultra-sonication. The soluble protein was separated from the cell debris by centrifugation at 12,000 rpm for 30 min at 4 °C. Other strains were expressed by the same strategy.

## Purification and isolation of DoxA, FDX1 and FDR2

After centrifugation, the supernatant was loaded onto a column containing HisPur™ Ni-NTA Resin (GE Healthcare) for His-tag affinity purification. The column was washed five times with wash buffer (50 mM Tris, 300 mM NaCl, 20% glycerol, 50 mM imidazole, pH 7.5) to remove contaminating proteins. The target protein was eluted with elution buffer (50 mM Tris, 300 mM NaCl, 20% glycerol, 500 mM imidazole, pH 7.5). The finally obtained protein was analyzed using 12% sodium dodecyl sulfate polyacrylamide gel electrophoresis (SDS-PAGE).

FDR2 was further purified by size-exclusion chromatography at 10−20 °C on gel filtration column (GE Healthcare, Superdex 75), in three protein loads on a column. Then FDX1 was further purified by ion exchange column (GE Healthcare) with buffer A (20 mM MES, 20% glycerol, pH 6.5) and buffer B (20 mM MES, 1M NaCl, 20% glycerol, pH 6.5), in two proteins loads on a column. Purified fractions were checked on SDS-PAGE gel. Protein concentrations were measured with NanoDrop Spectrophotometer (Thermo Scientific) at 280 nm. Purified proteins were snap-frozen in liquid nitrogen and stored at −80 °C.

## CO-binding CYP assay

The reduced-CO difference spectrum of the DoxA was obtained according to Liu et al. (*Liu et al., 2003*). Briefly, the purified DoxA protein was reduced with a few sodium dithionites, and the sample was scan between 400 nm and 500 nm at room temperature. Finally, the sample cuvette was saturated with about 30 bubbles to 40 bubbles of CO at a rate of 1 bubble per second, and was scanned between 400 nm and 500 nm at room temperature. The concentration of active P450 was calculated as described by *Omura & Sato (1964)*.

## Pyridine hemochromagen assay for the determination of heme protein concentration

The determination of heme protein concentration was according to Barr and Guo (*Barr & Guo, 2015*). Solution I which contains 0.2 M NaOH, 40% (v/v) pyridine, 500 μM potassium ferricyanide ($K_3Fe(CN)_6$) was prepared in a tube. The solution of 0.5 M sodium dithionite

was prepared in 0.5 M NaOH in another tube. A cuvette containing 500 μL solution I and 500 μL buffer A was used as a reference for all absorbance measurements. 500 μL of purified protein in buffer A and 500 μL of the solution I were transferred to a cuvette and mixed well. The UV–Vis spectrum of the oxidized Fe III state was recorded immediately. To the cuvette was then added 10 μL of the sodium dithionite solution, and the UV–Vis spectrum of the reduced Fe II state was recorded immediately.

### Enzyme assay of DoxA

The activity of DoxA was assayed using the DNR (daunorubicin) substrate. The reaction mixture consisted of 6 mg mixed protein (DoxA+FDX1+FDR2), 50 μM glucose-6-phosphate, 0.5 U glucose-6-phosphate dehydrogenase, 200 μM cysteine, 5 μM NADPH, 5mM $MgCl_2$ and 100 μM DNR, and the reaction was carried out in 20 mM sodium phosphate buffer (pH 7.5). The reaction mixtures were incubated at 30 °C for 24 h, and the pH was adjusted to 8.0 to stop the reaction. The solution was subsequently freeze-dried and was diluted with methanol. The product was analyzed using high performance liquid chromatography (HPLC; Waters, Milford, MA, USA). The HPLC was performed under the following condition: Kromasil C18 (250 mm ×4.6 mm I.D.), composed of solvent A [pH 2.3, with TFA (v/v)] in water and solvent B (100% methanol), were used in a flow rate of 1.0 ml $min^{-1}$. Detection was carried out with a UV detector at 254 nm.

## RESULTS

### The expression of DoxA in pET22b and pET28a

In order to express DoxA protein (48 kDa), *doxA* gene was cloned into pET22b which contains C-$His_6$-tag and pET28a which contains N-$His_6$-tag. The C-$His_6$-tagged and N-$His_6$-tagged fusion protein DoxA could be expressed well (Fig. S3). However, the amount of protein seen as a band on an SDS-PAGE gel does not show how much of the protein is correctly folded and active (*Hussain & Ward, 2003*). Therefore, we need to detect whether the DoxA protein folds correctly.

As a P450 enzyme, the absorption peak of DoxA shifts from 420 nm to 450 nm when CO is added into the reducing agent, which can be used as a characteristic to detect the activity of DoxA. Therefore, evaluation of the levels of correctly folded recombinant P450 was carried out by determining the CO-reduced difference spectra (*Haudenschild et al., 2000*; *Simgen et al., 2000*). The purified DoxA was added with reducing agent sodium dithionite and CO was introduced, the absorbing form of DoxA was almost at 420 nm (Figs. 3A, 3B), which suggested DoxA fused with pET22b and pET28a may be folded incorrectly.

### Expression of DoxA in the presence of ferredoxin and ferredoxin reductase

Most CYPs require redox partner proteins to sequentially transfer two electrons form NAD(P)H to their heme–iron reactive center for dioxygen activation (*Sun et al., 2017*). The activity of correctly folded cytochrome P450s was further enhanced by cloning a ferredoxin reductase (*Hussain & Ward, 2003*). The study by *Rimal et al. (2015)* showed the most appropriate redox partners of DoxA are ferredoxin FDX1 (18 kDa) and ferredoxin

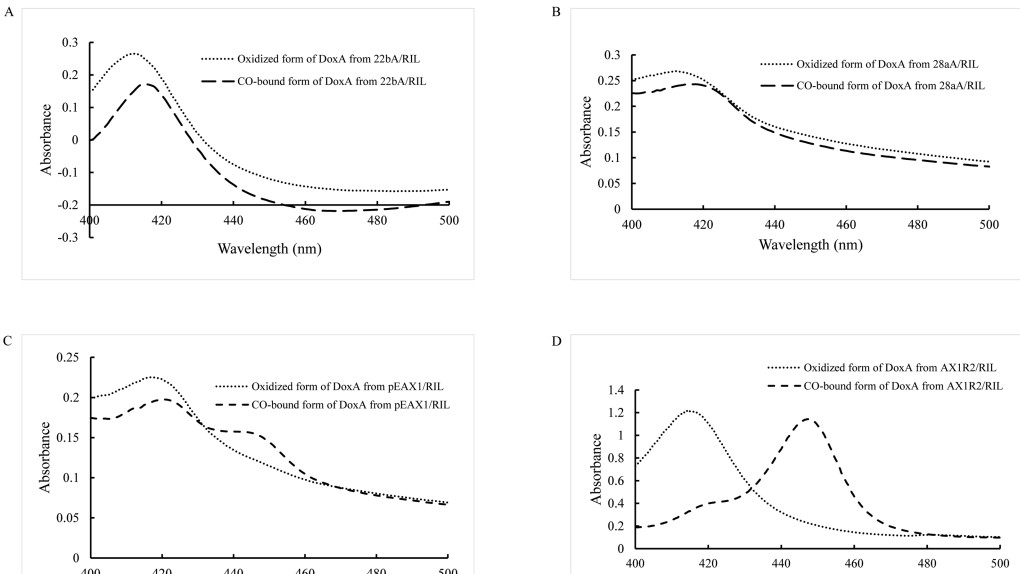

**Figure 3** CO-binding spectra of DoxA purified from 22bA/RIL (A), 28aA/RIL (B), pEAX1/RIL (C) and AX1R2/RIL (D).

**Table 3** The concentration of active P450 in different strains.

|  | 22bA/RIL | 28aA/RIL | pEAX1/RIL | AX1R2/RIL |
|---|---|---|---|---|
| Concentration of active P450 ($\mu$M)[a] | 0.1780 | 0.4286 | 0.7967 | 10.7956 |

**Notes.**
[a]Concentration of active P450 is calculated from the reduced-CO absorbance as described in Materials and Methods.

reductase FDR2 (55 kDa) in *S. peucetius* 27952. So, pEAX1/RIL which could co-express FDX1 and DoxA, AX1R2/RIL which could co-express FDX1, FDR2 and DoxA were constructed in this study, and CO-binding assay for the mixed protein was detected (Figs. 3C, 3D). The mixed protein DoxA and FDX1, and mixed protein DoxA, FDX1 and FDR2 were added with reducing agent sodium dithionite and CO was introduced, the absorption peak obviously shifted from 420 nm to 450 nm (Figs. 3C, 3D). The result showed that the active P450 forms of the mixed protein (DoxA, FDX1 and FDR2) were significantly more than that of *Rimal et al. (2015)*, and the highest P450 activity of DoxA was seen at pEAX1/RIL (Table 3). The CO-bound reduced difference spectra of mixed proteins showed the characteristic peak at 450 nm, confirming the expression of functional P450 enzymes. It is suggested that DoxA can be folded correctly with the help of FDX1 and FDR2 to ensure its activity.

## The determination of heme protein concentration for DoxA

The catalytic activity of P450s requires one or more redox partners to transfer two electrons from NAD(P)H to the heme iron. As a heme-containing protein, the heme concentration is also an important factor to detect the folding of DoxA. According to *Barr & Guo (2015)*,

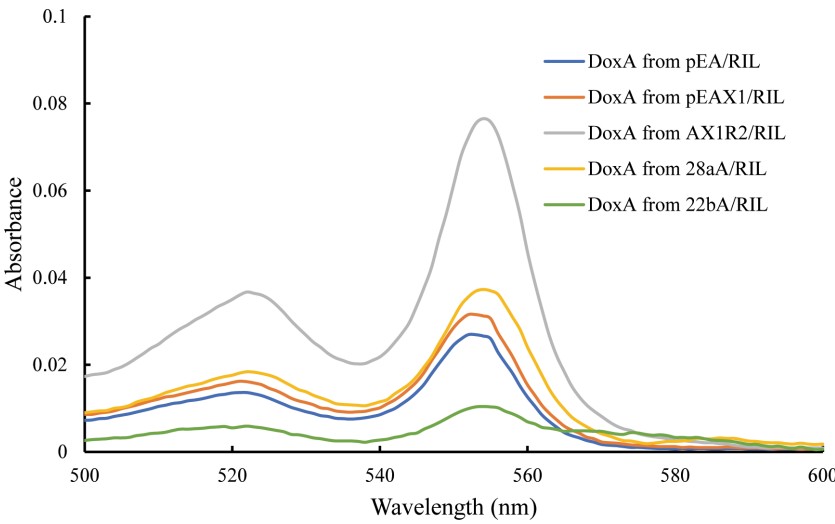

**Figure 4** Determination for the heme concentration of DoxA in different expression strains.

the heme concentration of DoxA expressed in different vectors were detected, and the concentration of DoxA and mixed protein from different plasmids is keep the same. We observed the heme concentration of DoxA expressed with FDX1 and FDR2 was higher than that expressed with FDX1 and expressed alone (Fig. 4), which indicated the activity of DoxA can be enhanced with the help of FDX1 and FDR2.

## Isolation of DoxA for the co-expression strain AX1R2/RIL

DoxA, FDX1 and FDR2 were successfully purified together by Ni-NTA Resin (Fig. 5B, Lane 1), and the protein DoxA was further isolated by size-exclusion chromatography and ion exchange chromatography (buffer A: 20mM MES, 20% glycerol, pH 6.5; buffer B: 20 mM MES, 1M NaCl, 20% glycerol, pH 6.5) (Fig. 5). The result showed FDX1 was isolated by ion exchange chromatography, and FDR2 could be isolated by size-exclusion chromatography. As a result of cell culture 2.7 g/L of wet weight cell was obtained and DoxA was purified with a yield of 0.725% by Ni-NTA, 28.12% by size-exclusion chromatography, and 19.48% by ion exchange chromatography (Table 4).

## The bioconversion of DNR to DXR by DoxA

In order to determine the activity of DoxA, the DNR was used as substrate and the reaction product was analyzed by HPLC. The product was eluted with the same retention time (17.60 min) as authentic DXR, while there was no corresponding peak in the control (Fig. 6), which suggests DNR could be converted to DXR by DoxA using the redox partner FDX1/FDR2.

## DISCUSSION

DoxA belongs to CYP129A family, which contains only three proteins: DoxA from *S. peucetius* 27952, DoxA from *S. peucetius* 29050 and DoxA from *S. peucetius* C5. DoxA
A
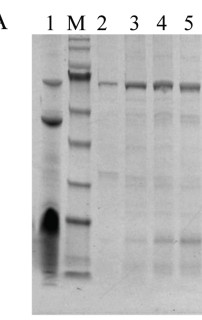
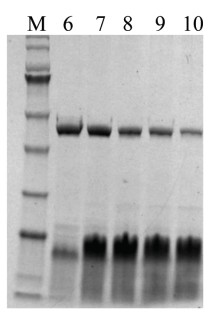

B
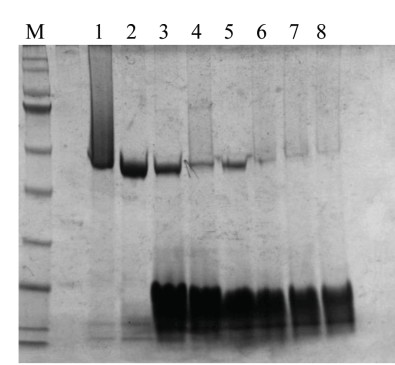

**Figure 5** **The purification and isolation of DoxA.** (A), SDS-PAGE for the elution of size-exclusion chromatography. Lane 1, DoxA, FDX1 and FDR2 were purified together by Ni-NTA Resin; Lane 2–5, FDR2 was isolated by size-exclusion chromatography; Lane 6–10, DoxA and FDX1 were eluted together by size-exclusion chromatography; (B), SDS-PAGE for the elution of ion exchange chromatography. Lane 1–2, DoxA was isolated by ion exchange chromatography; Lane 7–8, FDX1 was isolated by ion exchange chromatography; M, protein marker (Genestar, 10, 15, 25, 35, 45, 65, 75, 100, 135 and 180 kDa).

**Table 4** **Summary of the DoxA purification.**

| Step | Total protein (mg)[a] | Target protein (mg)[b] | Purity (%)[c] |
|---|---|---|---|
| Ni-NTA | 6734.6 | 48.9[*] | 0.725 |
| Size-exclusion chromatography | 50.9 | 14.3 | 28.12 |
| Ion exchange chromatography | 7.9 | 1.5 | 19.48 |

Notes.
[a] Protein concentration determined by Bradford assay using BSA as a standard protein.
[b] Determined from total protein concentration and purity.
[c] Purity determined by densitometric assessment of SDS-PAGE.
[*] Recombinant protein.

can catalyze continuous multi-step oxidation reactions on different carbon atoms, and the number of this kind of P450 oxidase is relatively few. The P450 oxidase MycG (PDB: 2YCA) involved in the biosynthesis of mycinamicin catalyzes hydroxylation and also epoxidation at C-14 and C-12/13 on the macrolactone ring of mycinamicin (*Anzai et al., 2012*; *Li et al., 2012*). Aurh (PDB: 3P3Z) involved in the biosynthesis of aureothin catalyzes the hydroxylation and oxidation to form the aureothin tetrahydrofuran ring (*He, Müller & Hertweck, 2004*; *Zocher et al., 2011*). Chle2 involved in chlorotricin biosynthesis can catalyze the multi-step oxidation of methyl groups at the same position of the substrate through hydroxyl, aldehyde and carboxyl groups (*Jia et al., 2006*). A cytochrome P450 protein, FkbD, catalyzes a less common, four-electron oxidation at C-9 to give a rarely found $\alpha$-keto amide group (*Chen et al., 2013*). Sequence alignment with these P450 enzymes showed that there is highly conserved heme bound Cys, EXXR motif in K-helix and Thr in I-helix in DoxA. There are also conserved iron porphyrin binding site F-(SGNH)-X-(GD)-X-(RHPT)-X-C-(LIVMFAP)-(GAD) and $O_2$ binding site (GA)-G-X-(DE)-T (Fig. 7).

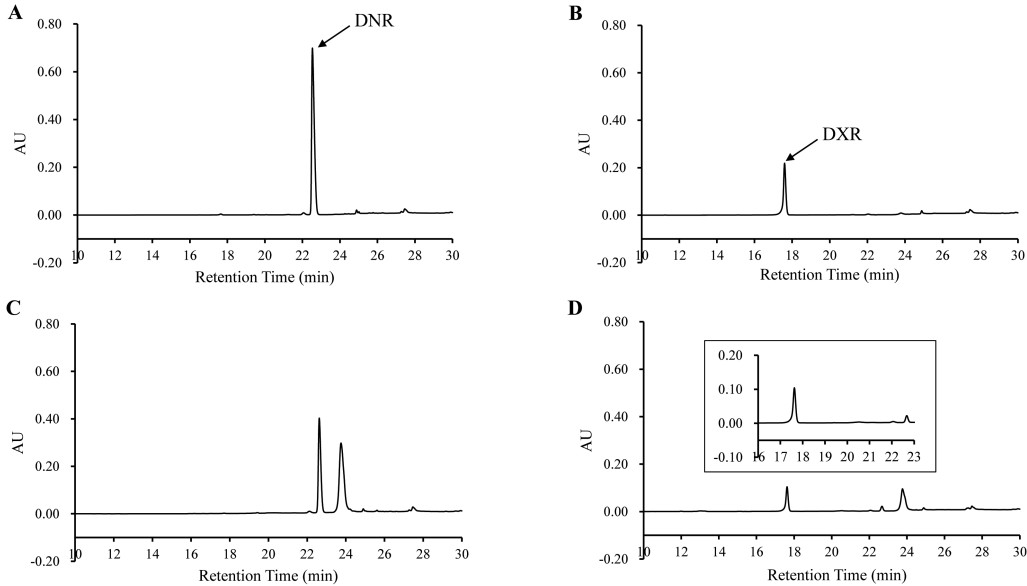

**Figure 6  HPLC chromatograph of DoxA reactions.** The peaks denote the DXR production. (A), DNR standard (100 μM); (B), DXR standard (100 μM) ; (C), Reaction control; D, Reaction of DoxA with FDX1-FDR2. The reaction mixture consisted of 6 mg mixed protein (DoxA+FDX1+FDR2), 50 μM glucose-6-phosphate, 0.5 U glucose-6-phosphate dehydrogenase, 200 μM cysteine, 5 μM NADPH, 5mM MgCl$_2$ and 100 μM DNR, and the reaction was carried out in 20 mM sodium phosphate buffer (pH 7.5). The reaction mixtures were incubated at 30 °C for 24 h. While the control reaction does not contain proteins.

The cytochrome *doxA* genes have been expressed previously in *E. coli*, and it is mostly in inactive P420 forms (Fig. S1, *Rimal et al., 2015*). Therefore, we attempted to develop an efficient expression system for DoxA to allow production of active, correctly folded enzyme. We firstly tried different expression vector to express DoxA alone, but the 450 nm reduced-CO difference spectrum was unable to observed (Fig. 3). This may be due to the improper protein folding or improper incorporation of the heme group into the apoenzyme in *E. coli*, just as the cytochrome P450 TxtC studied by (*Healy et al., 2002*).

Most CYPs require redox partner proteins to sequentially transfer two electrons form NAD(P)H to their heme–iron reactive center for dioxygen activation (*Sun et al., 2017*). Unlike monotonic eukaryotic cytochrome P450 reductases, bacterial redox partner systems are more diverse and complicated (*Li, Du & Bernhardt, 2020*). Although various orthologs of FDX and FDR are present in other bacterial strains, the heterologous reconstruction of the electron-transport partners in other host systems is often ineffective, suggesting that the employment of the most appropriate electron-transport partners is critical to obtain high CYP activity. According to *Hussain & Ward (2003)*, we co-expressed DoxA, ferredoxin and ferredoxin reductase. CO-reduced difference spectra and heme concentration showed that DoxA could be folded correctly when co-expressed with FDX and FDR, indicating that DoxA requires the presence of redox partners to perform its function normally. It appears that co-expressing the ferredoxin reductase with P450 and ferredoxin could stabilize the folded, active form of the P450. It may suggest that an *in vivo* association of these proteins

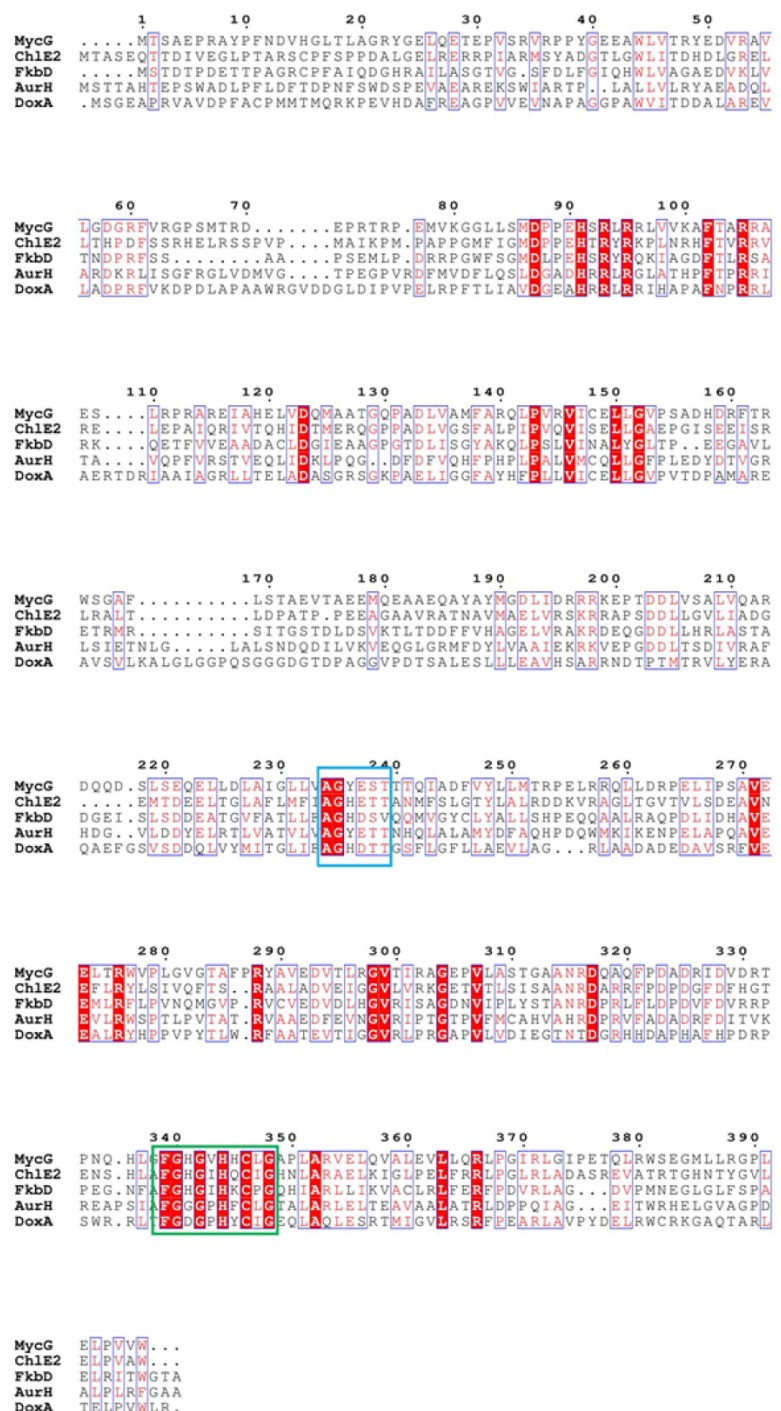

**Figure 7** **The homologous sequence alignment of DoxA.** The blue box is the O$_2$ binding site of P450 protein, and the green box is the iron porphyrin binding site.

can stabilize the P450 (*Hussain & Ward, 2003*). It could be applied to the enhancement of other cloned P450 enzymes.

In view of the highest concentration of active p450 in the mixed protein purified from the pEAX1/RIL strain, we further isolated and purified DoxA in it. Size-exclusion chromatography is a partition chromatography that separates molecules according to their molecular sizes, so, we first used size-exclusion chromatography to separate DoxA, FDX1 and FDR2. The result showed FDR2 could be isolated, while DoxA and FDX1 could not be separated by size-exclusion chromatography (Fig. 5). This made us try ion exchange chromatography to further separate DoxA and FDX1, which is a method to separate proteins according to the different charges of proteins under certain pH conditions. The protein DoxA was finally isolated by ion exchange chromatography. Recently, our group tried to crystallize DoxA and analyze its structure for a better understanding of the protein, which can be helpful in the study of similar types of CYP monooxygenases.

The efficiency of the electron transport pathway in a bacterial Class I P450 system is determined by mutual interactions of five elements including P450, Fdx, FdR, substrate, and NADPH (*Zhang et al., 2018*). It is known that different redox partners can change the product distribution of a P450-catalyzed reaction (*Guo et al., 2021*). There are six FDXs and seven FDRs in *S. peucetius* 27952 (*Rimal et al., 2015*), FDX1-FDR2 was used as the redox partner of DoxA for enzyme activity experiment in our study, and the main product DXR was generated in the reaction (Fig. 6, Fig. S4). We also carried out reaction with other redox partners including 0978FDR/1499FDX (*Zhang et al., 2018*) and spinach FDR/FDX, and DXR was not detected using HPLC, but there were other products in these reactions (Fig. S5), which will be studied later.

## CONCLUSION

Overall, our work has demonstrated that it is possible to optimize construct design and expression system to generate the soluble and active DoxA. These approaches should be applicable to other P450s. The success in DoxA expression and purification will facilitate future structural studies to understand how DoxA carries out the final hydroxylation for conversion of DNR to DXR, and to reveal the molecular basis of the catalysis and exquisite substrate specificity of DoxA.

We are grateful to Dr. Shengying Li for kindly providing the materials 0978FDR/1499FDX used in this research.

### Funding

This work was supported by the Natural Science Foundation of Guangxi Province (Grant No. 2018GXNSFAA281019 and 2022GXNSFBA035536), and the National Natural Science Foundation of China (Grant No. 31860245 and 31960203). The funders had no role in study design, data collection and analysis, decision to publish, or preparation of the manuscript.

## Grant Disclosures

The following grant information was disclosed by the authors:

Natural Science Foundation of Guangxi Province: 2018GXNSFAA281019, 2022GXNSFBA035536.

National Natural Science Foundation of China: 31860245, 31960203.

## Competing Interests

The authors declare there are no competing interests.

## Author Contributions

- Liyan Yang conceived and designed the experiments, performed the experiments, analyzed the data, prepared figures and/or tables, authored or reviewed drafts of the article, and approved the final draft.
- Dengfeng Yang performed the experiments, prepared figures and/or tables, and approved the final draft.
- Qingyan Wang performed the experiments, prepared figures and/or tables, and approved the final draft.
- Juan Li performed the experiments, prepared figures and/or tables, and approved the final draft.
- Hong-Liang Li performed the experiments, prepared figures and/or tables, and approved the final draft.
- Lixia Pan conceived and designed the experiments, analyzed the data, authored or reviewed drafts of the article, and approved the final draft.

## DNA Deposition

The following information was supplied regarding the deposition of DNA sequences:

The doxA sequence is available at GenBank: CP022438.1.

## Data Availability

The raw data is available in the Supplementary Files.

## Supplemental Information

Supplemental information for this article can be found online at http://dx.doi.org/10.7717/peerj.14373#supplemental-information.

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
