# Peer review of "Functional expression and purification of DoxA, a key cytochrome P450 from Streptomyces peucetius ATCC 27952"

_PeerJ, doi:10.7717/peerj.14373_

## Round 0.1 · original submission · Major Revisions

Before I can send your paper to reviewers, I must ask you to perform several changes:

A) The purification table (Table 3) is hard to interpret and seems to be wrong. i found there the following issues:
1) how was the amount of target protein determined?
2) how is it possible that almost 90% total protein in the crude extract comes from your overexpressed construct?
3) If indeed 90% of total protein in the crude extract is your target protein, how did the high-affinity Ni-NTA column fail to capture most of it? This makes no sense at all!
4) 48.9 mg over 6743.6 mg is not 7.25%, but 0.725%
5) Why are no data regarding total enzyme activity collected in each stept provided? Those are standard data required for protein prurification tables, which usually contain also a column depicting the specific activity to enable the computation of purification factors

B) in fig.2, please highlight the portion of the molecule where the changes between daunorubicin and doxorubicin occur
C) in the legend to Fig. 6, please describe the composition of "control" , as well as the reaction conditions (pH, time, tmeperature, protein concentration, etc.) used to oobtain the "reacted" sample.

D) The paper does not include information on the specific activity of the purified sample, nor indeed sufficient information to ascertain if it contains any information above that shown by other groups' attempts at over-expression of this protein in an active form (e.g. 10.1128/JB.181.1.298-304.1999 10.1016/j.abb.2015.08.019). This must be remedied.

---

## Round 0.2 · Major Revisions

Please address the issues highlighted by our reviewers.

Reviewer 1 ·

Basic reporting

1. The molecular weight of DoxA, FDX1 and FDR2 should be repoted in the manuscript.
2. In Figure 7 the proteins of ChlE2 and FkbD have not introduced in the manuscript.
3. There is no scale bar to display the concentration of DNR and DXR.

Experimental design

1. The authors should give some explains why to use size-exclusion chromatogrphy and ion exchange chromatography to purify the proteins of DoxA, FDX1 and FDR2.
2. In the pyridine hemochromagen assy for the determination of heme protein concentration whether the concentration of DoxA from different plasmids is keep the same.

Validity of the findings

1. the proteins of DoxA, FDX1 and FDR2.are expresssed in the same bacteria seperately but not as fusion protein. Why the authors can stated that DoxA can folded correctedly with the help of FDX1 and FDR2?
2. The CO-spectra of DoxA in pET22b and pET28a shows at 420nm. It is because of the Cys residus in DoxA are oxided. It is not as that it is not folded incorrectly.

Additional comments

1. The molecular weight of DoxA, FDX1 and FDR2 should be repoted in the manuscript.
2. In Figure 7 the proteins of ChlE2 and FkbD have not introduced in the manuscript.
3. There is no scale bar to display the concentration of DNR and DXR.
4. The authors should give some explains why to use size-exclusion chromatogrphy and ion exchange chromatography to purify the proteins of DoxA, FDX1 and FDR2.
5.. In the pyridine hemochromagen assy for the determination of heme protein concentration whether the concentration of DoxA from different plasmids is keep the same.
6. the proteins of DoxA, FDX1 and FDR2.are expresssed in the same bacteria seperately but not as fusion protein. Why the authors can stated that DoxA can folded correctedly with the help of FDX1 and FDR2?
7. The CO-spectra of DoxA in pET22b and pET28a shows at 420nm. It is because of the Cys residus in DoxA are oxided. It is not as that it is not folded incorrectly.

Reviewer 2 ·

Basic reporting

The written English of this article is not clear, unambiguous or technically correct. The article must be rewritten in professional languages.

The research background in the introduction was not sufficient. Please provide more details about “DoxA has been purified by Rimal et al., but it is mostly in inactive P420 forms”.

“The unprecedented high-efficiency and functional expression and purification of DoxA in E. coli expression system was realized.” Please explain how efficient was the expression and purification?

Experimental design

no comment

Validity of the findings

Figure 5, generally, there is no need to provide size-exclusion chromatography of proteins in the article.

No conclusions are well stated and the impact and novelty of the study was not assessed in the discussion section.

Reviewer 3 ·

Basic reporting

no comment

Experimental design

no comment

Validity of the findings

no comment

Additional comments

In this manuscript, a key cytochrome P450 DoxA from Streptomyces peucetius was functional expressed and purified. The high-efficiency and functional expression and purification of DoxA in E. coli expression system was realized. This research has practical significance for scalable production of doxorubicin. I recommend that this manuscript can be published in the journal. Moreover, some revisions are required. There are some issues to be addressed.
1. In the ‘Enzyme assay of Dox’ part, please give the full name of the abbreviation of
DNR.
2. Table 3, the authors had better provide the more Purification parameters such as activity, specific activity and yield.
3. In this manuscript, some of the contents in the results section belong to Materials and methods and discussion. The results section needs to be simplified. In addition, I suggest that the authors combine the results part with the discussion part.
4. The bioconversion of DNR to DXR by DoxA: For the verification of the transformation product, the authors should provide further confirmation data of the structure such as MS or NMR.

---

## Round 0.3 · accepted · Accept

All reviewers confirm that their comments have been addressed. This manuscript is ready for publication.

Reviewer 1 ·

Basic reporting

no comment

Experimental design

no comment

Validity of the findings

no comment

Additional comments

The authors have revised the manuscript according to the Reviewers' suggestions. I recommend to its publication in its revised form.

Reviewer 2 ·

Basic reporting

no comment

Experimental design

no comment

Validity of the findings

no comment

Additional comments

The authors showed that the mixed protein (DoxA, FDX1 and FDR2) purified from pEAX1/RIL was almost in active P450 forms,which were significantly more than that of Rimal et al. And the concentration of active P450 of DoxA in different expression strains was calculated, the highest P450 activity of DoxA was seen at pEAX1/RIL. The novelty of the study was added in the discussion section. Questions were addressed.

Reviewer 3 ·

Basic reporting

The authors has made serious revisions according to the comments of reviewers. I recommend that this manuscript can be published in Peer J.

Experimental design

no comment

Validity of the findings

no comment

Additional comments

The authors has made serious revisions according to the comments of reviewers. I recommend that this manuscript can be published in Peer J.